# High-Temperature Ferroelectric Behavior of Al_0.7_Sc_0.3_N

**DOI:** 10.3390/mi13060887

**Published:** 2022-05-31

**Authors:** Daniel Drury, Keisuke Yazawa, Andriy Zakutayev, Brendan Hanrahan, Geoff Brennecka

**Affiliations:** 1Colorado School of Mines, 1500 Illinois Ave., Golden, CO 80401, USA; keisuke.yazawa@nrel.gov; 2National Renewable Energy Laboratory, 15013 Denver West Parkway, Golden, CO 80401, USA; andriy.zakutayev@nrel.gov; 3U.S. Army Combat Capabilities Development Command—Army Research Laboratory, Adelphi, MD 20783, USA; brendan.m.hanrahan.civ@army.mil

**Keywords:** AlScN, ferroelectric, high temperature, nonvolatile memory, retention, fatigue, wurtzite, film, sputter deposition

## Abstract

Currently, there is a lack of nonvolatile memory (NVM) technology that can operate continuously at temperatures > 200 °C. While ferroelectric NVM has previously demonstrated long polarization retention and >10^13^ read/write cycles at room temperature, the largest hurdle comes at higher temperatures for conventional perovskite ferroelectrics. Here, we demonstrate how AlScN can enable high-temperature (>200 °C) nonvolatile memory. The c-axis textured thin films were prepared via reactive radiofrequency magnetron sputtering onto a highly textured Pt (111) surface. Photolithographically defined Pt top electrodes completed the capacitor stack, which was tested in a high temperature vacuum probe station up to 400 °C. Polarization–electric field hysteresis loops between 23 and 400 °C reveal minimal changes in the remanent polarization values, while the coercive field decreased from 4.3 MV/cm to 2.6 MV/cm. Even at 400 °C, the polarization retention exhibited negligible loss for up to 1000 s, demonstrating promise for potential nonvolatile memory capable of high−temperature operation. Fatigue behavior also showed a moderate dependence on operating temperature, but the mechanisms of degradation require additional study.

## 1. Introduction

Nonvolatile memories, which do not require a permanent/frequent voltage supplied to maintain the bit state, are important for storing information indefinitely and reliably. The concept of high−operating−temperature nonvolatile memory (HOT−NVM) has been an elusive capability. Current computational and data storage limitations in harsh environments either require on−board cooling or locating sensors/computations away from the heat source. These harsh environments for electronics are a focus of the NASA HOTTech project and can be found within jet turbines, within deep−well drilling, and on the surface of Venus [1].

Nonvolatile random−access memory (RAM) based on FLASH, magnetic, phase−change, and resistive mechanisms degrade quickly even at moderate temperatures (<200 °C) [2,3,4,5]. Microelectromechanical (MEM) and nanogap resistance switching (NGS) offer promise for elevated temperatures NVM, but there are downsides with moving parts and operating in various atmospheres [6,7]. Ferroelectric technology based on perovskite or fluorite structures (e.g., Pb(Zr,Ti)O_3_ or (Hf,Zr,Si)O_2_) is currently limited to temperatures < 200 °C [8,9]. This is a result of a destabilized polar structure, chemical instabilities, and increased domain wall mobility in Pb(Zr,Ti)O_3_ [10,11], whereas increased pyroelectric contributions and depolarization fields plague (Hf,Zr)O_2_ compositions when increasing temperature [12,13]. Here, we demonstrate promising initial results when using ferroelectric Al_0.7_Sc_0.3_N to meet the HOT−NVM demands.

Since its discovery in 2019 as the first wurtzite nitride ferroelectric, Al_1__−__x_Sc_x_N (AlScN) has received much attention from both the ferroelectric and piezoelectric communities [14]. The high remanent polarization (P_r_) associated with the ideal wurtzite structure is an attractive property for increasing either device density or extracted charge density. Previous studies on AlN−based ferroelectrics reported the high coercive field (E_c_) necessary to switch between the two polarization states. While this high E_c_ is problematic for integration with low−voltage devices, it can be valuable for retention characteristics as temperatures increase [8]. The E_c_ of AlScN can be tuned through various modifications such as composition [14,15,16], stress [17], epitaxial alignment [18], and temperature [19,20,21,22]. In addition to thickness scaling, these trends are important for integrating AlScN into a high−temperature chip to reduce the operating voltages.

To assess polarization stability for a nonvolatile memory, Islam et al. poled the sample at room temperature (RT), baked the package at 1100 °C, and remeasured the P_r_ once cooled back to RT [23]. While promising for an NVM since the polarization state is maintained after exposure to extreme temperatures, these ex situ results did not address the P_r_ or switching behavior at elevated temperatures. Liu et al. focused on the retention state at RT for an AlScN ferroelectric−gated field effect transistor (FET) device by monitoring changes in the resistance state over time [24]. Furthermore, an NVM must endure switching cycles without compromising the P_r_ or E_c_ values, which can impact deciphering between either polarization state, and the device must avoid electrical shorts. Two recent reports have noted that AlScN has low endurance, a measure of repeated bit flips, at <10^5^ cycles before dielectric failure, which may be limited by the slim margin between coercive and breakdown fields (E_bd_) for the samples, but a true mechanistic understanding of fatigue in AlScN is still lacking [25,26]. Thus, these reports do not represent a fundamental limitation of the ferroelectric switching endurance. In this article, we report device relevant behavior (i.e., leakage, retention, and fatigue) between 23 and 400 °C to introduce AlScN as a candidate material for HOT−NVM.

## 2. Materials and Methods

The 225 nm thick ferroelectric AlScN film was deposited by radiofrequency (rf) reactive magnetron sputtering onto a highly textured Pt (111) bottom electrode. The base pressure of the chamber reached 2 × 10^−6^ Torr prior to flowing the deposition gases. The pressure was maintained at 2 mTorr while flowing 5 sccm N_2_ and 15 sccm Ar. Furthermore, 300 W of rf power was supplied to the 5 cm diameter Al_0.7_Sc_0.3_ alloy target. The substrate was heated to 400 °C and positioned 16.5 cm from the target. Details of sputter deposition processes using in situ optical emission spectrometry were previously described [27]. The target was conditioned with a 1 h pre−sputter to remove surface oxidation and stabilize the reactive target mode. Following the nitride layer deposition, lithographically defined 50 μm diameter platinum top electrodes were sputter−deposited to form a metal−ferroelectric−metal (MFM) structure for electrical testing.

The resultant AlScN film was structurally characterized by X−ray diffraction (XRD) using Cu Kα radiation. A Panalytical X’Pert MRD diffractometer was used to scan the *θ*−2*θ* and *ω* space. The sample was tested in a high−temperature vacuum probe station to mitigate possible oxidation of the nitride. Electrical properties of the MFM capacitors were measured using a ferroelectric analyzer (Radiant Precision Premier II).

The MFM capacitors were evaluated with DC current−voltage (IV) sweeps to assess the resistivity at various temperatures; this is important for evaluating polarization values that may be inflated by charge leakage. Polarization−electric field (P–E) hysteresis loops were collected between probe station temperatures of 23−400 °C using an excitation frequency of 10 kHz. This frequency was chosen to reduce the leakage contribution to the P_r_ values, and 10 kHz is a relatively high frequency when compared to previous reports on ferroelectric AlScN, which range between 333 Hz and 100 kHz [19,21]. Retention testing was used to measure the stability of a polarization state over time to inspect the volatility of that state. The test involved a positive or negative pre−pulse to the bottom electrode to pole the sample in either a metal or a nitrogen polar orientation, respectively. This was followed by a dwell for a length of time, and then the MFM capacitor was pulsed twice with the same voltage profile (Figure 1). The measured 2P_r_ elucidates whether the capacitor underwent ferroelectric polarization switching between pulse 1 and pulse 2 or not. In this study, four different pulse sequences were necessary to characterize the same state and opposite state retention characteristics for each polarization state—nitrogen (N) and metal (M) polar. A ‘same state’ (SS) test indicates that the bias of the read and write pulses is the same, while an ‘opposite state’ (OS) test switches pulse polarity. The difference between a switched and unswitched polarization response is 2P_r_. If the 2P_r_ shrinks over time, then the bit state will eventually be indecipherable; therefore, a stable polarization value is essential for NVM memory applications. Ten virgin capacitors with an active (top electrode) diameter of 50 μm were assessed at each temperature. P–E loops (10 kHz) were collected prior to each 1000 cycle (1 kHz triangular waveform applied for 1 s) fatigue to inspect the ferroelectric properties throughout the process.

## 3. Results and Discussion

### 3.1. Structural and Leakage Characteristics

In Figure 2a, the XRD scan reveals a wurtzite phase of AlScN with c−axis texture. The (0002) reflection appeared at 36.056° in *2θ* with a full width at half maximum (FWHM) value of 0.158°. The ω−rocking scan of the (0002) position resulted in an FWHM value of 2.694°, which is comparable to prior reports (Figure 2b) [16,22]. Lack of additional wurtzite peaks associated with other orientations indicates that the film had a purely c−axis texture. Since the c−axis is the polar axis of the wurtzite structure, it is important to maximize the degree of c−axis out of plane texture when investigating the ferroelectric properties of an MFM capacitor (inset of Figure 2a).

The P–E loops measured at 10 kHz between 23 and 400 °C reveal a strong dependence of the E_c_ on temperature (Figure 3a). The square−like hysteresis loops were a result of a relatively low permittivity, defined energy barriers between polarization states, and minimal loss mechanisms. Levels of polarization saturation similar to previous reports were reached at each temperature [14,16]. At 400 °C, the derivative dP/dE became negative after saturation, indicating decreased capacitor resistance. This observation is supported by the current density–electric field (J–E) sweeps in Figure 3b. When subjected to a 1 MV·cm^−1^ DC field, the current density rose from 1 × 10^−7^ A·cm^−2^ at 23 °C to 3 × 10^−4^ A·cm^−2^ at 400 °C. Furthermore, J increased exponentially with E and maintained this relationship at all temperatures. The increasing leakage with temperature is important for compensation when possibly measuring changes in polarization (e.g., during retention testing).

The ΔE_c_/2 decreased linearly as temperature increased, describing a reduced activation barrier due to thermal energy contribution. An E_c_ temperature coefficient of 4.5 kV·cm^−1^·K^−1^ describes our results in Figure 3c, which showed a decrease from 4.3 to 2.6 MV·cm^−1^ between 23 and 400 °C. In the limited number of reports that analyzed the E_c_ and P_r_ dependence on temperature for Al_1__−x_Sc_x_N, there were a range of compositions studied with various frequencies. Our data (30% Sc, 10 kHz) showed a slightly lower E_c_ temperature coefficient in comparison to the literature. The overall E_c_ results (apart from the reduced E_c_ values reported by Wang et al.) are consistent with the literature in terms of a lower E_c_ due to increasing Sc content or decreasing excitation frequency. This may explain both the coincidental values between Gund et al. (30% Sc, 3 kHz) and Zhu et al. (16% Sc, 333 Hz) and the increased E_c_ for Mizutani et al. (20% Sc, 100 kHz) [19,20,21,22]. Our work shows that the P_r_ values for Al_0.7_Sc_0.3_N remained relatively stable up to 400 °C (Figure 3d). The temperature−independent P_r_ behavior of an Al_1__−__x_Sc_x_N composition was also noted by the mentioned reports. In general, the P_r_ decrease with increasing *x* was seen in a majority of the reports [16]. 

### 3.2. Polarization Retention at Elevated Temperatures

Retention studies were carried out relative to the surface polar state, nitrogen or metal, termed SS_N_, SS_M_, OS_N_, and OS_M_. Polarization changes were investigated with durations of pulses up to 1000 s to reveal if any changes occurred. A reduction in measured polarization would indicate that an internal process occurred, reducing the net polarization via mechanisms such as back−switching, domain pinning, or phase degradation. However, our results revealed no P_r_ loss, indicating a kinetically stable polarization direction and a negligible change in the polarization−pinning defect environment (Figure 4). This contrasts perovskite and fluorite−based ferroelectric MFM capacitors, which showed a clear reduction in P_r_ when subjected to a bake between the write and read voltage pulses [9,12,28]. For both SS tests, there was a marginal level of polarization (<6 μC·cm^−2^), which may have been the effect of changes in the leakage contribution between the first and second read pulses. Since the OS and SS tests for the metal and nitrogen poled devices showed a steady polarization retention between 1 and 1000 s, the estimated time until the 2P_r_ was reduced enough for the bit state to become indecipherable requires further retention testing before a reasonable estimate can be established. Our results are also in agreement with the RT retention tests performed by Fichtner et al., who reported steady retention up to 10^5^ s [14]. Overall, the ferroelectric AlScN retained all the polarization that was written into the MFM capacitor, even while operating at 400 °C. This characteristic is a promising factor for enabling high−temperature nonvolatile memory.

### 3.3. Device Fatigue at Elevated Temperatures

To ascertain the switching endurance of the sample, 10 MFM capacitors from the same sample were tested at each temperature. The maximum field applied during the fatigue cycles and P–E loop measurements was scaled to 105% of the E_c_. Figure 5a provides a summary of the fatigue results, which were also temperature−dependent. A P–E loop measurement followed every 1000 cycles to track the changes in the polarization throughout the fatigue subjection. One standard deviation is denoted by the error bars in Figure 5a. Device failure was defined at the point when the device shorted. The highest average endurance occurred at 200 °C (2.7 × 10^5^ ± 1.9 × 10^5^ cycles) with a maximum of 6.0 × 10^5^ cycles. While this is an increased fatigue endurance relative to previous reports on AlScN, it still indicates that effort is needed to increase the margin between E_c_ and E_bd_ and to investigate and mitigate the underlying causes of fatigue in these materials. Because E_c_ decreased with increased temperature faster than E_bd_ decreased, the increased margin between the two was presumably an important contributor to the increase in fatigue life from RT to 200 °C. The degraded performance at temperatures > 200 °C, however, was associated with larger leakage currents in the MFM capacitor that occurred during each fatigue cycle.

The least change in polarization versus cycling happened at 100 °C. Figure 5b–f are representative snapshots of the P–E loop measured after the given number of cycles. Especially for the higher−temperature panels, the initially square−like hysteresis loops deteriorated into a combination of lower P_r_ and inflated polarization values due to an uncompensated leakage contribution. Additional study into the cycle−dependent leakage characteristics may point toward a specific leakage mechanism that increases in effect with cycling. Furthermore, a systematic investigation on the AlScN switching fatigue behavior for various electrode types will be critical for applications that require more resilient endurance.

## 4. Conclusions

In summary, a thin film, c−axis textured AlScN was electrically tested in a metal–ferroelectric–metal capacitor for polarization hysteresis, retention, and fatigue up to 400 °C. The leakage current density increased by four orders of magnitude between RT and 400 °C, which revealed the importance of compensating for the leakage contribution to polarization, especially as temperature increased. While a strong E_c_ dependence on temperature (4.5 kV·cm^−1^·K^−1^) led to a 60% reduction at 400 °C compared to RT, the P_r_ remained stable over all temperatures. As an important characteristic for NVM, in situ polarization retention testing up to 1000 s revealed negligible (<2%) polarization loss for either the metal or the nitrogen polar state at all temperatures tested up to 400 °C. On average, the highest endurance before complete failure occurred at 200 °C with 2.7 × 10^5^ switching cycles; however, the reduction in P_r_ versus cycles was lowest at 100 °C. Our results showing >10^5^ switching cycles are encouraging for using ferroelectric AlScN in HOT−NVM applications where <10^3^ switching cycles would be required. Future work on the fatigue mechanisms at different temperatures and expanding the retention duration time are vital avenues for understanding the intrinsic limitations of AlScN.

## Figures and Tables

**Figure 1 micromachines-13-00887-f001:**
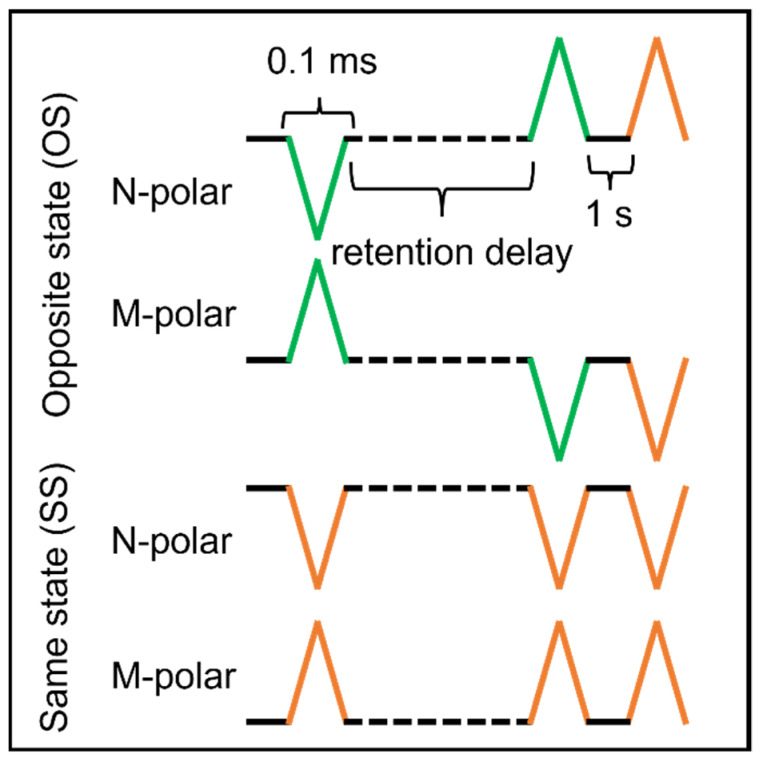
Retention test protocols for OS and SS in either a nitrogen or metal polar surface state. Green and orange traces represent switching and non−switching pulses, respectively.

**Figure 2 micromachines-13-00887-f002:**
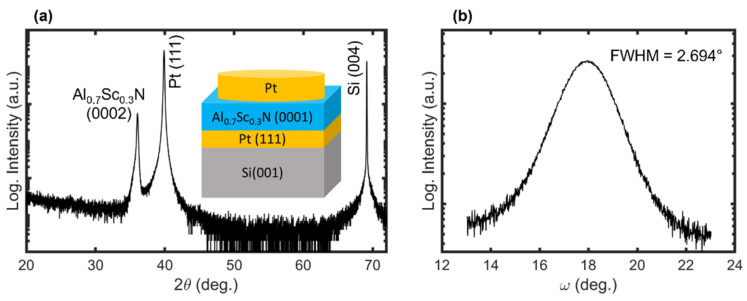
(**a**) The *2θ–θ* XRD scan of the investigated sample revealing a pure wurtzite phase with c−axis texture. The inset depicts the stacking sequence of the metal–ferroelectric–metal capacitor. (**b**) *ω* rocking scan of the Al_0.7_Sc_0.3_N (0002) peak.

**Figure 3 micromachines-13-00887-f003:**
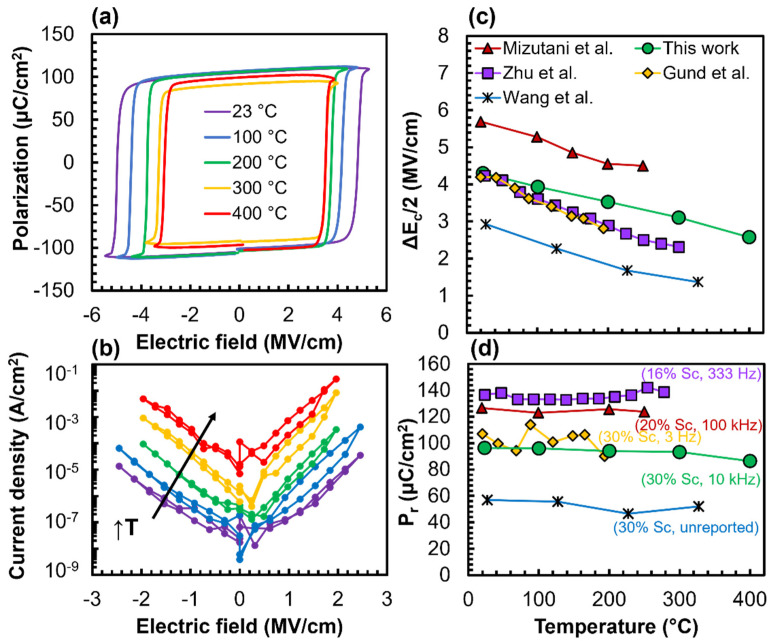
(**a**) P−E loops between 23 and 400 °C collected with 10 kHz bipolar voltage waveform. (**b**) Temperature−dependent leakage current density (J) over bias between 23 and 400 °C. Comparison of (**c**) ∆E_c_/2 and (**d**) P_r_ results from this work with Mizutani et al. [21], Zhu et al. [20], Gund et al. [22], and Wang et al. [20]. The results in (**a**,**b**) are from this work, while (**c**,**d**) are a comparison of our results with previous reports.

**Figure 4 micromachines-13-00887-f004:**
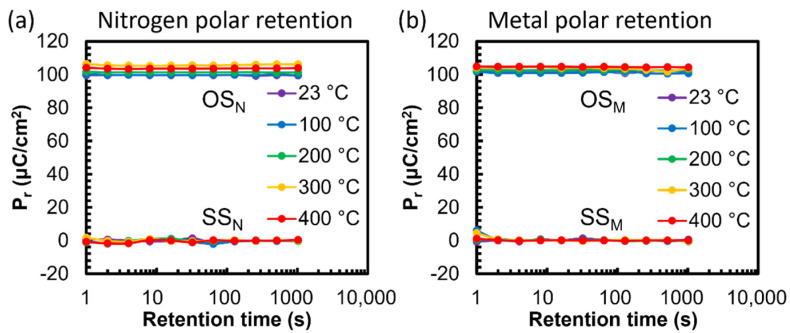
Polarization retention after poling in either (**a**) nitrogen or (**b**) metal surface state and dwelling up to 1000 s at the indicated temperature.

**Figure 5 micromachines-13-00887-f005:**
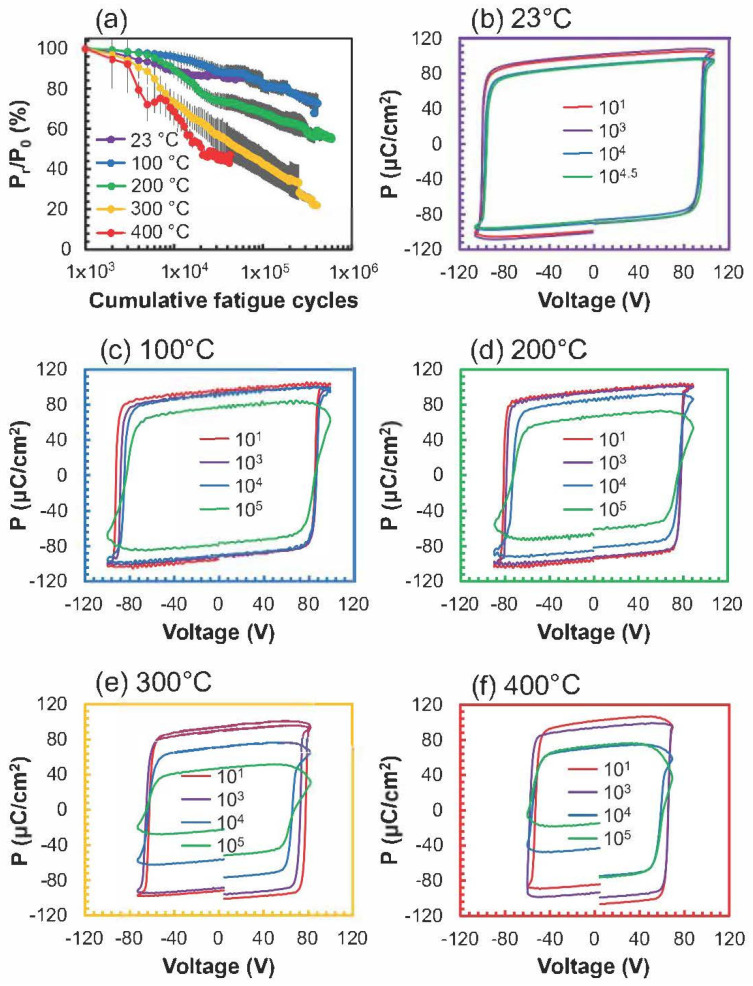
Ferroelectric fatigue exhibits a strong dependence on the measurement temperature. (**a**) Normalized P_r_ dependence on cycles for temperatures between 23 and 400 °C with error bars of a standard deviation. Representative P–E loops throughout the fatigue cycling for (**b**) 23 °C, (**c**) 100 °C, (**d**) 200 °C, (**e**) 300 °C, and (**f**) 400 °C. The colors of the plot borders for (**b**–**f**) correspond to the temperature of the measurement in (**a**), and the different traces in (**b**–**f**) are consistent for ease of comparison.

## Data Availability

The data that support the findings of this study are available from the corresponding author upon reasonable request.

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
