# Peer review of "High-Temperature Ferroelectric Behavior of Al0.7Sc0.3N"

_micromachines, 2022, doi:10.3390/mi13060887_

Round 1

Reviewer 1 Report

I reviewed with interest the manuscript titled “High temperature ferroelectric behavior of Al0.7Sc0.3N”. The authors report measurements of ferroelectric loops, leakage current, polarization retention and fatigue of c-oriented AlScN thin films in the temperature range between RT and 400 °C. The authors interpret the results as encouraging for high operating temperature non-volatile memories.

The paper is well written, original, and presents results that are of significant interest for the scientific and engineering community. After updating the paper following (minor) comments and questions below I agree with its publication in the journal Micromachines.

Introduction

1What is expected Tc of ferroelectric AlScN material?

Materials and Methods

2 What is thickness of the films?

3Rocking curve has FWHM of 2.694 °. Can the authors benchmark this value to the existing data on c-AlScN films?

4The polarization loops were measured at 10 kHz, which is fairly high. What is the reason for this?

   Figure 5: Can the authors explain in the caption different colouring used for the curves? Also, can they explain further in text what are N-polar and M-polar states and how to distinguish between the two? To someone experienced in oxide ferroelectrics this is not obvious.

Results

The films break down before 10^6 cycles and the authors suggest that further studies are needed to mitigate this behaviour. Nevertheless, could the authors discuss potential influence of the electrode? Why way Pt chosen as top/bottom electrode? Would choosing other (TiN, for instance) influence the behaviour?

Reviewer 2 Report

Very nice paper on high temperature AlScN for application as non-volatile memory. The paper is clearly written and scientifically sound. Improvement should be made on presentation of diagrams: 

- Figure 2: lettering of insert too small

- Figure 4: use same scale for both diagrams

- Figure 5: lettering too small (too many diagrams, use two diagrams per line)

These abbreviations need to been introduced: HZO, FWHM.

Field emission scanning electron microscope has been introduced but no images of microstructure shown. It would be nice to have some pictures.

Please check notation of units: space character after number and before, when symbols in advance. Use 10x instead of 2E-6 Torr.

Sentence in line 206/207 not clear.

In Conclusions chapter the decrease in resistivity has been pointed out but it hasn't been shown or discussed in the Results and Discussion chapter. Where do the values come from?
